# Psychological and Physiological Responses in Patients with Generalized Anxiety Disorder: The Use of Acute Exercise and Virtual Reality Environment

**DOI:** 10.3390/ijerph17134855

**Published:** 2020-07-06

**Authors:** Tsai-Chiao Wang, Cindy Hui-Ping Sit, Ta-Wei Tang, Chia-Liang Tsai

**Affiliations:** 1Institute of Physical Education, Health & Leisure Studies, National Cheng Kung University, Tainan 701, Taiwan; chiao.ellen@gmail.com; 2Department of Sports Science and Physical Education, The Chinese University of Hong Kong, Hong Kong; sithp@cuhk.edu.hk; 3Department of Leisure and Recreation Management, Asia University, Taichung 413, Taiwan; 4Department of Medical Research, China Medical University Hospital, Taichung 413, Taiwan; 5Institute of Innovation and Circular Economy, Asia University, Taichung 413, Taiwan

**Keywords:** virtual environment, acute exercise, EEG, perceived stress, restorative quality, satisfaction, generalized anxiety disorder

## Abstract

Virtual exercise therapy is considered a useful method by which to encourage patients with generalized anxiety disorder (GAD) to engage in aerobic exercise in order to reduce stress. This study was intended to explore the psychological and physiological responses of patients with GAD after cycling in a virtual environment containing natural images. Seventy-seven participants with GAD were recruited in the present study and randomly assigned to a virtual nature (VN) or a virtual abstract painting (VAP) group. Their electroencephalogram alpha activity, perceived stress, and levels of restorative quality and satisfaction were assessed at baseline and after an acute bout of 20 min of moderate-intensity aerobic exercise. The results showed that both the VN and VAP groups showed significantly higher alpha activity post-exercise as compared to pre-exercise. The VN group relative to the VAP group exhibited higher levels of stress-relief, restorative quality, and personal satisfaction. These findings imply that a virtual exercise environment is an effective way to induce a relaxing effect in patients with GAD. However, they exhibited more positive psychological responses when exercising in such an environment with natural landscapes.

## 1. Introduction

Generalized anxiety disorder (GAD) is one of the most common mental disorders. It is characterized by persistent, invasive, excessive problems that make daily life difficult [1,2]. Patients with GAD have pre-existing anxiety, nervous reactions (e.g., anxiety, tremors, and headaches) or worry about many events or activities, which will result in their feeling highly stressed and finding it difficult to relax [3,4]. Accordingly, GAD reduces their quality of life and interpersonal relationships owing to the negative effects of stress [5]. If patients with GAD are not properly treated with effective stress-relief strategies, they will succumb to depression or panic disorder in the long run. 

Virtual environment (VE) technology has the potential to provide people with a high degree of immersion and presence, which may make individuals with high stress loads feel that they have escaped from the actual surrounding environment [6]. The Cave VE surrounds users by projecting stereoscopic images on a large screen, thereby providing them with a highly immersive experience. In particular, the Cave projection-based VE system allows users to interact with the environment, which in turn allows those who are exercising indoors to feel as if they are exercising in an outdoor environment (e.g., with forest and grass) [7]. Such an isolated environment is thus considered to be a useful tool for promoting indoor and outdoor physical activities (e.g., [7,8]). The characteristics of this virtual reality are important for patients with GAD [9,10] since they are afraid of coming into contact with people. VE can immerse them in a pleasant situation in which to engage in exercise without having to engage with others [11,12].

An important challenge while using Cave VE in improving the quality of life of patients with GAD is to select a suitable virtual scenario, ensure that it has a relaxing effect, and is able to facilitate their engaging in exercise. Stress reduction theory is an important framework explaining why contact with nature might foster stress reduction [13,14] through a relaxing effect on the parasympathetic nervous system [15,16]. This theory argues that stress occurs when individuals encounter events or situations that are perceived to be unfavorable, threatening, or challenging to them [16]. The natural environment has been proposed to facilitate recovery from physiological stress [13,14,17]. Contact with natural places will produce a relatively fast (within minutes) affective reaction at a subconscious level that can be measured through physiological pathways [14,16,18] where exposure to natural settings initiates an innate, rapid, affect-driven process that reduces physiological and psychological stress [16,17]. Thompson et al. (2011) [19] compared the physical and mental health of individuals who participated in physical activities in natural outdoor environments with those engaging in physical activity indoors. They found that exercising in natural environments was associated with more feelings of revitalization and positive engagement, thus lessening tension and depression more than exercising indoors. 

Individuals can perform physical activities (e.g., walking, running, riding a bicycle) in simulated landscapes such as forests, parks, woods, and rivers produced by machines, where they can feel immersed in the natural environment through a VE [7,20,21]. Similar to exercising in a natural environment, individuals exercising in simulated virtual natural environments may not only obtain benefits related to their physical health [7], but also may obtain benefits including reducing perceived stress and obtaining restorative effects [22]. Previous studies have reported that cycling in a virtual natural environment can meet individual’s needs for exercise and stress relief, in cases where they are satisfied with exercise in such an environment [16,23]. However, user’s perceptions of value have a critical impact on future behavior [24]. The value perspective argues that providing users with higher value (i.e., reducing stress and perceived recovery) is a key to obtaining higher levels of satisfaction [25,26]. Indeed, provision of a virtual exercise environment is not the only function of such devices; they also provide a sensory experience, with the former referring to the ability to promote engaging in physical activity and the latter referring to the extent to which they feel positive psychological effects during exercise. 

At present, many indoor exercisers use a simple static environment, but such a scenario makes users feel bored [27]. The advantages of the virtual environment are that it could create a more attractive indoor exercise environment. Therefore, the present study uses virtual abstract painting (VAP) as an alternative approach to compare the psychological and physiological responses induced by a virtual nature environment. Abstract painting is one of the main categories of visual art. While observing the natural environment can promote relaxation [14,16], observing fine art can also lead to pleasant feelings [28]. The advantage of abstract paintings is that they lack objective visual content, thus allowing participants to interpret the images according to their own preferences [29]. Accordingly, when exercising in this environment, individuals will not exert excessive amounts of attention and can feel good without being bored at the same time.

To understand an individual's current psychological response state, it is possible to measure changes in the reflex potential produced by the activity of nerve cells in the brain [30]. It is known that alpha waves are related to a relaxed state [30,31]; however, when mental stress or workload is reduced, alpha waves increase [32]. Indeed, Hassan et al. (2018) [31] found that participants observing photos of natural landscapes experiences enhanced brain electrical activity in the form of alpha waves.

To the best of our knowledge, no research has yet been conducted on the potential effects of exercise in a virtual environment with various types of environmental stimuli on psychological and physiological responses in patients with GAD. As virtual reality exercises become more and more popular [20,33] and VE is considered to be a useful tool for promoting exercise and physical health in patients with GAD (e.g., [7]), it is important to understand the psychophysiological responses of patients with GAD (e.g., levels of relaxation, perceived stress, restorative effects, and satisfaction) while exercising in a VE [34,35] since they are important factors related to their subsequent usage [24]. Although a previous study has reported that exercising in a natural environment may benefit physiological and psychological health [22], outdoor exercise and indoor virtual exercise seem to induce different psychological benefits, with greater energy being experienced while exercising outside whereas more relaxation and less tension is felt while exercising inside with virtual reality environment [36]. Valtchanov et al. (2010) [37] also suggested that immersion in virtual nature environments could produce similar beneficial effects on restorative quality and stress relief as exposure to surrogate nature. Importantly, GAD is a psychiatric disorder characterized by fear and avoidance of most social situations and interpersonal relationships owing to the negative effects of stress [5,38]. Therefore, the purpose of this study was to explore the different effects of virtual reality exercises via an indoor Cave VE system comprising natural landscapes and abstract paintings on perceived stress, satisfaction, and levels of restorative effects in patients with GAD. We hypothesized that cycling in a VE would induce positive psychological responses; however, it was posited that more restorative effects and a higher degree of satisfaction could be produced by the natural landscapes compared to abstract paintings. To verify the proposed hypotheses, a randomized controlled trial was conducted.

## 2. Methods

### 2.1. Participants

The participants selected for this study were based on the following inclusion criteria: (a) an initial diagnosis of GAD based on the generalized anxiety disorder 7–item (GAD-7) scale owing to its high sensitivity and specificity to detect GAD [2,39]; (b) aged between 50 and 75 years because exercise leads to several benefits (e.g., reduced mortality rate, delayed cognitive aging, and lowered medical costs) for middle-aged and older adults [40,41,42]; (c) normal body mass index (BMI) (18.5 ≤ BMI <24 kg/m^2^) as defined by the Taiwan Ministry of Health according to the related morbidity data and mortality risks for Asian populations [43,44]. Exclusion criteria were that the participants had the following: (a) obsessive compulsive disorder or other anxiety disorder; (b) a mini mental state examination (MMSE) score less than 24, representing cognitive impairment; and (c) suffering from claustrophobia since the experiment was performed in a narrow, immersive surround system. Eighty-four participants were contacted, and seven participants were excluded because they were diagnosed with other anxiety disorders by a physician. The study was approved by the Human Research Ethics Committee of the National Cheng Kung University (B-ER-107-150) in Taiwan. Written informed consent was obtained from all participants, in accordance with the Declaration of Helsinki.

A power analysis (G*Power 3.1.9.4) was used to calculate the sample size required for conducting the survey to obtain at least a small-to-medium effect (*r* = 0.20), using an alpha-level of 0.05 (two-tailed) [45]. Power was set at 0.80 [46]. The results of the G*Power analysis suggested a required sample size of N = 36. Thus, the sample size (*n* = 77) used in this study was suitable for examining the hypotheses.

### 2.2. Procedures

This experiment required participants to concentrate for approximately an hour during the experiment. In order to lower potential risk and confounding factors that would interfere with physiological and psychological responses, prior to the experiment the researchers alerted the GAD patients within 24 hours by phone to avoid unwanted behaviors (e.g., staying up late, drinking caffeinated beverages, and taking medications). 

Seventy-seven GAD patients diagnosed and referred to by the physician were randomly assigned to the virtual nature (VN, *n* = 40) or the virtual abstract painting (VAP, *n* = 37) group. Each participant was asked to arrive at the laboratory at about 8:30–9:30 am to control for circadian influences. When they arrived at the laboratory, the research assistant explained the experimental procedure, and the participant was asked to complete an informed consent form, a demographic questionnaire, the MMSE, the GAD-7, and perceived stress questionnaires. Their height and weight were also measured to calculate their body mass index (BMI). Then, electroencephalogram (EEG) and heart rate (HR) were measured to ascertain that there was no difference between their relaxed and emotional state before the intervention.

Both the VN and VAP groups cycled 20 min in a Cave VE (see Figure 1). The Polar optical HR sensor worn on the participant's arm was used to monitor their HR during cycling. All participants were asked to exercise at a moderate intensity of 50-60% HRmax. in the VN group, landscapes of forests, parks, woods, and rivers generated by machine simulations were projected in the Cave and moved as the participants stepped onto the bicycle. In the VAP group, a slideshow of abstract paintings was projected in the Cave, with each painting appearing for one minute in random order. These abstract paintings were selected without special meaning for the participants. The paintings present flowing images in common colors (such as green, blue, and yellow). After the exercise intervention, they underwent an EEG exam and completed the questionnaires (see Figure 2).

### 2.3. Virtual Environment

The Cave VE system, which was installed in an acoustically shielded room with dimmed lights, projected the content around the participants, enhancing the sense of immersion and presence and allowing them to feel that they were cycling outdoors [21,47]. An environment was established that allowed him/her to engage in exercise in a wraparound environment. The VE was presented in a surround view within the participant's visual field. The wraparound screen of the VE appeared in front of the participant and to the left, right, and underneath. The bicycle was placed two meters from the front screen. Using wraparound 3D projection technology to project images onto multiple large screens to render images, the entire scene could be projected to form a surround scene [7,21]. The hardware components of the Cave virtual reality system were mainly the projector and the screen. For the surround projection virtual reality, two projectors were used to project images at 270 degrees, and the participant could see the display of the 3D natural environment content with the naked eye. This system uses wireless serial technology to synchronize the cycling speed along with the flow image speed in the VE (Combined virtual reality device, Patent No. I67522, Taiwan). A similar Cave virtual reality system was successfully used in previous studies (e.g., [7,47]). All of the experimental steps were performed in the VE Laboratory at National Cheng Kung University in Taiwan. The temperature during the experiment was controlled at 24–25 °C, and the relative humidity was controlled between 50–60%.

### 2.4. Psychological Response Assessments

Three questionnaires were assessed with a 5-point Likert scale (1 = “strongly disagree”; 5 = “strongly agree”) in the present study.

#### 2.4.1. Restorative Quality

The restorative quality scale was adapted from four questions developed by Van den Berg et al. (2016) [48]. The scale items were “cycling in the virtual environment makes me feel relaxed”; “cycling in the virtual environment makes me feel good”; “cycling in the virtual environment is appealing to me”; “the virtual environment looks beautiful.” The reliability of the restorative quality scale in the individuals with GAD was determined using the Cronbach's alpha coefficient before the formal experiment, with a coefficient of 0.93 indicating satisfactory internal consistency.

#### 2.4.2. Perceived Stress

The level of perceptual stress was measured using a short version of the perceived stress scale developed by Cohen et al. (1983) [49]. It comprises a total of seven items. The reliability of the perceived stress scale in the individuals with GAD was determined using the Cronbach's alpha coefficient before the formal experiment, with a coefficient of 0.91 indicating satisfactory internal consistency. A higher perceptual pressure score represented lower levels of perceived pressure.

#### 2.4.3. Satisfaction

Satisfaction was measured by the four questions developed by Oliver (1997) [50] and Cronin et al. (2000) [51]. The scale items were “I am satisfied with cycling in this virtual exercise environment”; “I am happy to cycle in this virtual exercise environment”; “I am pleased to have made the decision to cycle in this virtual exercise environment”; “Overall, I am satisfied with cycling in this virtual exercise environment.” The reliability of the satisfaction questionnaire in the individuals with GAD was determined using the Cronbach's alpha coefficient before the formal experiment, with a coefficient of 0.87 indicating satisfactory internal consistency.

### 2.5. EEG Collection and Processing

Stress prior to the intervention was measured through a variety of physiological EEG signals. The ProComp Infiniti biofeedback system (Thought Technology Ltd., Montreal, Canada) was used for data acquisition. EEG signals were continuously recorded through biosensors placed on the participants.

Variations in the activity of the neurons in the brain cause fluctuations in the voltage potential along the scalp that can be measured with an EEG signal [52]. Previous research has identified a number of brain wave frequency bands from EEG data, including delta, theta, alpha, and beta waves. Alpha waves (8–13 Hz) can typically be observed when an individual is in a relaxed state. The higher the value of the alpha wave is, the more relaxed the individual is [53,54]. 

In the present study, the EEG signals were recorded from four channels (FP1, FP2, T3 and T4) and placed on each participant’s scalp according to the international 10-20 system. The mean power from the average of all four electrodes was used [8,55]. Before the exercise intervention, two and a half minutes of resting EEG activity were recorded. The peripheral signals (e.g., blinking and muscle activity) were filtered by a moving average filter to remove noise. Electrode impedances were generally maintained below 5 kOhms. Prior to digitization, a low pass filter was set at 60 Hz, with a continuous sampling rate of 256Hz (The ProComp Infiniti biofeedback system, Thought Technology Ltd., Canada). The EEG data was filtered using a band pass filter in a frequency band of 0.2–0.35 Hz. 

### 2.6. Data Analysis

SPSS version 21.0 (SPSS Inc., IBM, Chicago, IL, USA) was utilized for all of the statistical analyses. The descriptive statistics of the data were expressed as mean ± SD (see Table 1). An independent sample t-test was used for between-group demographic comparisons. EEG alpha values and perceived stress scores were separately submitted to a 2 (*Group*: VN vs. VAP) × 2 (*Time*: pre- vs. post-intervention) repeated-measures analysis of variance (RM ANOVA). Posterior comparisons of the mean values were performed in the form of paired multiple comparisons using the Bonferroni correction where a significant difference occurred. A *p* value less than 0.05 was considered statistically significant. Cohen’s d was used as an appropriate assessment of effect size for significant t-test results [56], with a value of 0–0.2 for smaller effect sizes, 0.2–0.5 for medium effect sizes, and 0.5–0.8 for larger effect sizes [46].

## 3. Results

### 3.1. Demographic Characteristics

No participant reported any discomfort throughout the 20 min experiment. There were no significant differences in demographic variables between the VN and VAP groups before the intervention (Table 1).

### 3.2. Physiological Index-EEG Alpha Wave

As shown in Figure 3, the RM ANOVA on the EEG alpha value showed a significant effect of *Time* [*F*(1,75)  =  38.01, *p* < 0.001, *η_p_*^2^ = 0.34] and *Group* [*F*(1, 75) = 19.22, *p* < 0.001, *η_p_*^2^ = 0.21], indicating that post-exercise alpha values (4.77 ± 0.22 μV) were higher than pre-exercise values (3.15 ± 0.15 μV) across the two groups, and the alpha values for the VN group (4.52 ± 0.18 μV) were higher than those for the VAP group (3.39 ± 0.19 μV) across the two time points. The main effect was superseded by the *Time* × *Group* [*F*(1, 75) = 16.89, *p* < 0.001, *η_p_*^2^ = 0.19] interaction (see Figure 3). The post-hoc analyses indicated that the post-exercise alpha values were higher than the pre-exercise values for the VN [pre-exercise vs. post-exercise: 3.17 ± 1.38 μV vs. 5.87 ± 2.28 μV; *p* < 0.001] and VAP [pre-exercise vs. post-exercise: 3.12 ± 1.19 μV vs. 3.66 ± 1.33 μV; *p* = 0.01] groups.

### 3.3. Psychological Indices

#### 3.3.1. Perceived Stress

The RM ANOVA on the perceived stress scores revealed a significant main effect of *Time* [*F*(1,75)  =  30.97, *p* < 0.001, *η_p_*^2^ = 0.30) and *Group* [*F*(1, 75) = 6.63, *p* < 0.001, *η_p_*^2^ = 0.15], indicating that the post-exercise perceived stress scores (4.1 ± 0.07) were higher than the pre-exercise values (3.40 ± 0.10) across the two groups, and the perceived stress scores for the VN group (3.88 ± 0.09) were higher than those for the VAP group (3.58 ± 0.09) across the two time points. The main effect was superseded by the *Time* × *Group* [*F*(1, 75) = 12.76, *p* < 0.001, *η_p_*^2^ = 0.15] interaction. The post-hoc analyses indicated that the post-exercise perceived stress values were higher than the pre-exercise values only for the VN group (pre-exercise vs. post-exercise: 3.35 ± 0.78 score vs. 4.41 ± 0.52 score; *p* < 0.001) (See Figure 4), indicating that GAD patients in the VN group felt higher stress-relief levels.

#### 3.3.2. Restorative Quality and Satisfaction

As illustrated in Figure 5, statistically significant differences in restorative quality (*t* (1) = 7.67, *p* < 0.001) and satisfaction (*t* (1) = 6.86, *p* < 0.001) were observed between the VN and VAP groups. The VN group had significantly higher restorative quality levels as compared to the VAP group after the exercise intervention (VN vs. VAP: 4.07 ± 0.71 vs. 2.68 ± 0.88). The satisfaction level was also significantly greater in the VN group than in the VAP group after the exercise intervention (VN vs. VAP: 4.09 ± 0.71 vs. 2.98 ± 0.69). The effect sizes for the significant differences were large in the levels of restorative quality (*d* = 0.29)and satisfaction (*d* = 0.26). The main statistical findings are summarized in Table 2 and Table 3.

## 4. Discussion and Conclusion

The purpose of this study was to compare the different effects of virtual exercise environments on the levels of relaxation, perceived stress, restoration quality, and satisfaction in patients with GAD. We found that: (1) both the VN and VAP groups could obtain a significantly higher state of relaxation after exercising in the virtual environment; (2) patients with GAD felt higher levels of stress-relief and restorative quality, and were more satisfied when exercising via a Cave VE system with natural landscapes in contrast to abstract paintings. 

The virtual environment with natural landscapes has the characteristics of immersion and provides a presence effect [57,58,59]. The patients with GAD could experience more enjoyment and become more relaxed in such an environment. In addition, the virtual reality system simulated a situation of cycling outdoors, allowing the patients with GAD to have a front view that could be covered by the Cave virtual environment. Previous studies exploring the brain’s electrocortical responses to exercise have interpreted increased activity in alpha waves as an indicator of relaxation [60,61]. In the present study, the patients with GAD in both the VN and VAP groups showed higher EEG alpha activity after 20 min of cycling, suggesting that patients with GAD could feel more relaxed through exercise in an immersive virtual environment. This finding concurred with Cho’s (2017) [62], of a positive effect on the relative α-power spectrum after horseback riding. Similarly, Tsai et al. (2014) [63] also demonstrated that an acute bout of moderate exercise intensity could effectively lower cortisol levels and induce a relaxed state. Both the previous and present findings suggested that exercise with VE could be a suitable relaxation technique in patients with GAD regardless of the environment introduced. Therefore, VE can contribute to the relaxation of patients who are afraid to interact with others or with the outside world to engage in cycling at home or in indoor spaces.

However, only patients with GAD in the VN group relative to the VAP group showed significantly reduced perceived stress after an acute bout of 20-min of moderate cycling exercise. The present finding was partly in line with previous work trends suggesting that virtual exercise can reduce the level of stress perceived by users [64,65]. Furthermore, the stress reduction theory argues that environment can make a difference to stress levels [66]. Specifically, in the current study, the VN group obtained more benefits in terms of reduced stress than the VAP group when cycling in the virtual reality system. This finding also suggested that natural landscapes relative to abstract paintings are more suitable for releasing stress in patients with GAD, that the content presented in the virtual environment is particularly important for enjoying exercise, and that VR can affect the levels of stress perception in GAD patients. Also, a VE with natural landscapes can help patients with GAD gain the benefits of exercise while avoiding pressure related to social interaction. Indeed, Vyas et al. (2004) [67] found that exposure to nature can help patients manage and reduce the effects of anxiety and stress caused by chronic diseases. It is worth noting that a natural experience can reduce the burden of certain types of mental illness [13]. Accordingly, results of this study also suggest that cycling in a natural VE can reduce the burden of mental illness caused by GAD.

Compared to the virtual abstract paintings, the natural landscape shown by the virtual reality system benefited patients with GAD by contributing more restorative effects while they were cycling. Based on the stress reduction theory, previous research argues that natural environments provide more restorative benefits than indoor environments [22]. Shanahan et al. (2016) [22] recommended that the main goal of health policies is to establish a safe and restorative environment for recreation and various physical activities that in turn improve quality of life. The present study showed that simulated natural environments can help patients with GAD obtain higher levels of restorative effects. Consequently, the indoor virtual exercise environment with natural landscapes as compared with abstract paintings appeared to be a restorative environment to help patients with GAD recover their spirit and experience an enhanced quality of life. In addition, the present findings are consistent with those of previous studies. For example, Valtchanov et al. (2010) [37] examined whether immersion in virtual nature would produce a recovery effect and reported that participants immersed in a virtual natural environment gained more restorative effects as compared to the controls. Although previous studies have demonstrated that restorative quality could be facilitated by observing or exploring in a virtual nature environment (e.g., [37,59]) the present study further confirmed that exercise in such an environment can also lead to restorative effects in patients with GAD.

In addition, compared to the abstract painting images, the natural landscape generated by the wraparound VE allows individuals to be more satisfied with the cycling experience. The present finding concurs with previous findings reporting that the more people can reduce their stress or restore their spirit during an activity, the more satisfied they will be with the activity (e.g., [68,69]). Plante et al. (2007) [70] examined the psychological benefits of exercising in a natural environment compared with indoors and found that individuals who exercise outdoors in natural landscapes are more satisfied with their workout than those who exercise indoors. However, Hudson et al. (2019) [71] argued that, thus far, the impact of virtual environments on satisfaction has not been fully explored. More research on market positioning and communication about virtual experiences in other research contexts is warranted in this area, possibly examining virtual environments with different natural characteristics in terms of the relationships between satisfaction and marketing for patients with chronic diseases.

### 4.1. Limitations and Future Research

This study has several limitations that need to be considered when interpreting the findings. It only examined the effects of a single 20-minute virtual exercise period on psychological responses in patients with GAD, but did not examine the effects of long-term exercise. Acute aerobic exercise could only temporarily increase the levels of psychophysiological effects in healthy individuals [72] and patients with chronic disease [73], but long-term psychological and physiological benefits depend on regular exercise [74,75]. Therefore, to confirm the psychological responses of patients with GAD exercising in a virtual environment, in the future long-term intervention experiments should be conducted to determine the practical clinical benefits of virtual environment exercise for patients with GAD. Second, this study used a 270-angle virtual environment as a research tool rather than a 360-angle virtual environment, which can provide a higher degree of immersion. With the rapid development of technology, the virtual exercise experience will continue to improve. More research is needed in the future to explore the role of virtual environments in promoting exercise. Third, the psychological responses to an acute bout of exercise might be moderated by age [76,77]. Also, the effects of age on the virtual test of psychological performances were also reported in the previous study [78]. Since only middle-aged and older participants were recruited in the present study, further research is warranted in this area, possibly examining the effects of different virtual environments on psychological and physiological responses in young patients with GAD. 

### 4.2. Conclusions

Engaging in physical exercise in a natural environment has been demonstrated to lead to more benefits than when it is carried out in an indoor environment [19]. The quality of life for patients with GAD is susceptible to stress. Although cycling in both a natural VE environment and one in which abstract paintings are shown can lead to relaxation in patients with GAD, they perceived lower stress and higher levels of restoration quality and satisfaction when exercising in a natural VE environment. The present findings suggest that exercising in a virtual natural virtual environment is a feasible way to help solve clinical psychological and psychological problems and further improve the quality of life in patients with GAD.

## Figures and Tables

**Figure 1 ijerph-17-04855-f001:**
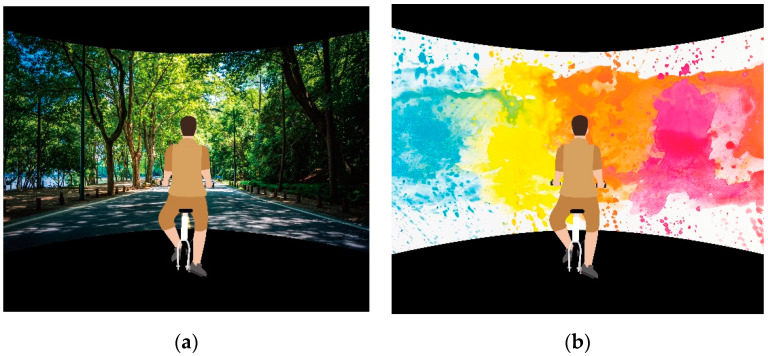
Experimental images. (**a**) virtual nature; (**b**) virtual abstract paintings (Experimental images from evening_tao | Freepik).

**Figure 2 ijerph-17-04855-f002:**
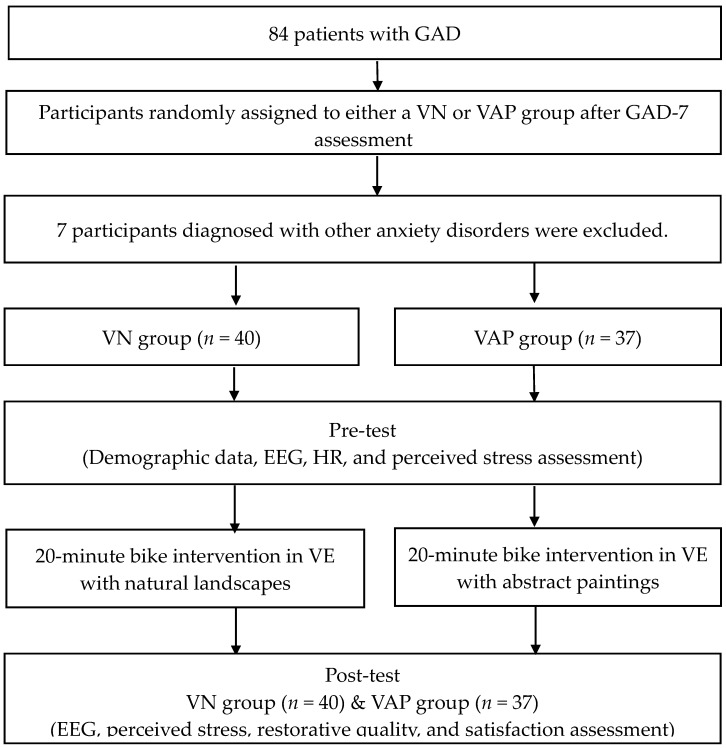
Flowchart of the study.

**Figure 3 ijerph-17-04855-f003:**
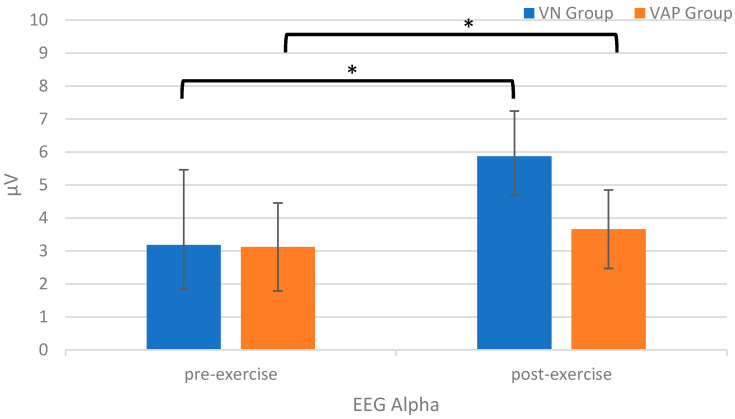
EEG alpha values (μV) for the virtual nature (VN) and the virtual abstract painting (VAP) groups before and after the exercise intervention (* *p* < 0.05).

**Figure 4 ijerph-17-04855-f004:**
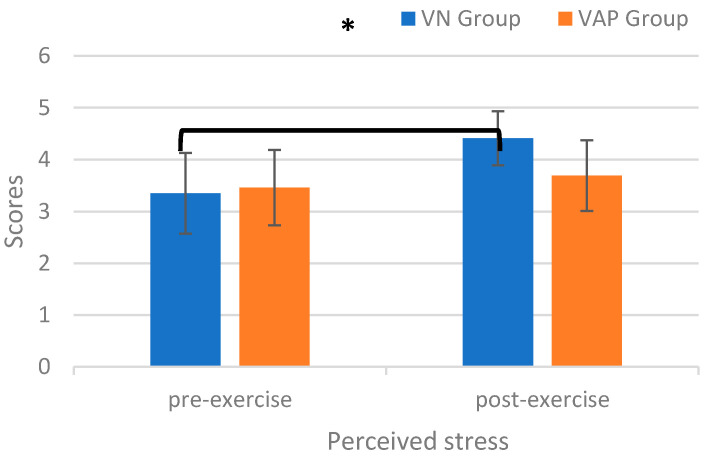
Perceived stress scores for the virtual nature (VN) and the virtual abstract painting (VAP) groups before and after the exercise intervention (* *p* < 0.05).

**Figure 5 ijerph-17-04855-f005:**
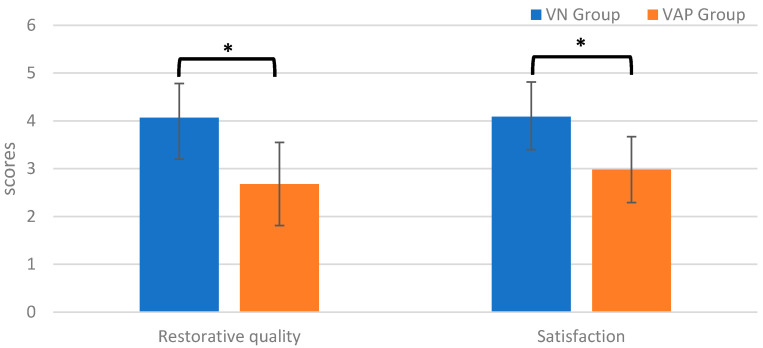
Restorative quality and satisfaction (Mean ± SE) for the virtual nature (VN) and the virtual abstract painting (VAP) groups after the exercise intervention (* *p* < 0.05).

**Table 1 ijerph-17-04855-t001:** Baseline demographic characteristics of the participants (mean (SD)).

Variables	VN Group (*n* = 40)	VAP Group (*n* = 37)	*p* Value
Age (year)	58.43 (7.37)	59.87 (6.99)	0.25
Gender (M/F)	21/19	18/19	0.47
Height (m)	1.60 (0.08)	1.60 (0.09)	0.91
Weight (kg)	60.22 (10.93)	61.97 (13.28)	0.53
BMI (kg/m^2^)	21.54 (1.25)	21.56 (1.77)	0.88
GAD levels (moderate/low)	28/12	24/13	0.47
MMSE (score)	28.81 (1.44)	29.13 (1.11)	0.29
GAD-7 (score)	12.43 (2.73)	12.73 (3.52)	0.78
Resting HR (count/minute)	77.91 (6.84)	79.83 (6.60)	0.46

Note: SD, standard deviation; BMI: body mass index; GAD: generalized anxiety disorder; MMSE: mini-mental state examination; HR: heart rate; VN: virtual nature; VAP: virtual abstract painting; HR: heart rate.

**Table 2 ijerph-17-04855-t002:** Comparisons of alpha value and perceived stress between the VN group and the VAP group.

Variables	Time	VN Group (*n* = 40)	VAP Group (*n* = 37)
Alpha value (μV)	Before cycling	3.17 ± 1.38	3.12 ± 1.19
	After cycling	5.87 ± 2.28	3.66 ± 1.33
	*p* Value	*p* < 0.001	*p* = 0.01
Perceived stress	Pre-exercise	3.35 ± 0.78	3.46 ± 0.93
	Post-exercise	4.41 ± 0.52	3.69 ± 0.64
	*p* Value	*p* < 0.001	*p* = 0.44

**Table 3 ijerph-17-04855-t003:** Comparisons of restorative quality and satisfaction levels between the VN group and the VAP group.

Variables	Time	VN Group (*n* = 40)	VAP Group (*n* = 37)	*p* Value
Restorative quality	Post-exercise	4.07 ± 0.71	2.68 ± 0.88	*p* < 0.001
Satisfaction	Post-exercise	4.09 ± 0.71	2.98 ± 0.69	*p* < 0.001

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
