# Peer review of "Psychological and Physiological Responses in Patients with Generalized Anxiety Disorder: The Use of Acute Exercise and Virtual Reality Environment"

_ijerph, 2020, doi:10.3390/ijerph17134855_

Round 1
Reviewer 1 Report
This is an interesting paper investigating new technology that could enhance treatment of anxiety disorders, GAD specifically. This paper seems worthy of publication, but I have a few questions and concerns:
Line 16-17 in the abstract is worded strangely and should be revised or removed: "However, this key issue has not been well understood, thus hindering the development of virtual exercise environments."
The introduction is well sourced and informative.
The inclusion of power analysis is appreciated in the methods section.
Why was a natural and/or indoor environment comparator not included? This would seem to be the standard way people exercise, it would be interesting to have the comparison. Please justify why this was not done.
In table 1, the BMI p-value seems surprisingly low given the group means and SD. Please check this.
Regarding the cohort characteristics, how well are the benefits observed in this study to generalize to younger cohorts?
For readers not familiar with average BMI in Taiwan, please place your observed BMI in context with the population.
Figures are fine, but more descriptive legends would improve the interpretation.
I would like to see a table organizing all of the main statistical findings in addition to the figures.
Please provide more justification for the use of Virtual Abstract Painting (VAP) as opposed to other potential comparison environments. Why not use a simple static environment that many users of aerobic exercise bikes would typically be exposed to?
Author Response
Please see the attachement.

Reviewer 2 Report
This article deals with a core future of GAD (stress and relaxation), and evaluates an innovative tool to reduce this problem. However, the manuscript has some limitations to consider.
Title and abstract
The title is unclear to me. What does “satisfy” mean in that context? In the abstract it is indicated that the goal of the study was “to explore the psychological responses of patients with GAD after cycling in a virtual environment with natural images”, thus the title does not fit with the aim of the research.
The abstract needs to be reviewed. Authors state that the goal of the study was to “to explore the psychological responses…” however, physiological variables were also studied. Moreover, other evaluated variables stated at the introduction (levels of relaxation, perceived stress, restoration quality, and satisfaction) are not clearly explained in the abstract. Furthermore, by reading the abstract only, I do not understand the meaning of “degree of satisfaction”. It is not clear if it is related with acceptability of the VR tool or personal satisfaction.
Furthermore, in the abstract results indicate that participants “exhibited more positive psychological responses when exercising in such an environment with natural landscapes” however, “only the VAP group exhibited significantly less perceived stress through exercise in the virtual environment.” I think these two assumptions need further clarification.
Introduction
In general, the introduction gathers a good review of the literature about the study variables. However, there are several points to consider:
On page 2, the authors explain the benefits of observing abstract painting (i.e. pleasant feelings) but they also state that observing natural environment promotes relaxation. On the other hand, the authors at the end of the introduction (page 3) hypothesize that “more restorative effects and a higher degree of satisfaction could be produced by the natural landscapes relative to abstract paintings.” I think the different or similar effects reported in the literature for both techniques have to be better justified and linked with the hypotheses.
Method
Participants:
One of the inclusion criteria was being physically fit but the authors do not explain the criteria used to evaluate physical fitness.
Procedure:
At the beginning of the procedure, the authors state that “This experiment required participants to concentrate for approximately an hour during the experiment. In order to avoid inattention on the part of the participants during the experiment, prior to the experiment, the researchers alerted them by phone to avoid behaviors (e.g., staying up late, drinking caffeinated beverages, and taking medications) within 24 hours that would interfere with physiological and psychological responses.” It seems that authors attempted to control concentration, however I do not think this procedure is enough to achieve it but just a way to reduce vulnerability.
Figure 1 does not indicate the number of participants that were evaluated at post-test.
Measures:
The measures are well described.
Results
The results section is well described although, in terms of format, I think it would be clearer to separate it by sections with subtitles.
Discussion
Overall authors captured the main study outcomes and described implications. Nonetheless, I have several considerations:
On pages 9-10, it is stated that “However, only patients with GAD in the VN relative to the VAP environments showed significantly reduced perceived stress after an acute bout of 20-min of moderate cycling exercise. The present finding was partly in line with previous work trends suggesting that virtual exercise can reduce the level of stress perceived by users [59,60]. However, the stress reduction theory argues that environment can make a difference in stress levels [61]. Specifically, in the current study, the VN group obtained more benefits in terms of reduced stress than the VAP group when cycling in the virtual reality system.”
This whole paragraph is not clear to me. Specifically, the results of VAP reducing perceived stress on one hand, and on the other hand, the VN group obtaining more benefits of reduced stress are not clear. These two points need further clarification.
In the conclusions section, the sentence "Since the quality of life for patients with GAD is susceptible to stress, the present findings suggest that exercising in a natural virtual environment is a feasible way to help solve this clinical problem.” does not fully explain the relationship between stress, quality of life and exercising in a natural environment.
Round 2
Reviewer 1 Report
I thank the authors for their revisions. I am mostly satisfied by the responses, but a few minor details remain:
I missed it on the first read, but there still seems to be a typo in Table 1, the p-value for weight is >1.
I like the new tables 2 and 3, but there seems to be a discrepancy in the sample size between the flow chart in Figure 2. Are there 37 or 40 participants in the VAP group? Or was the VN group just stated twice Figure 2? Please correct or explain.
Reviewer 2 Report
First, thank you very much for considering my comments. The new information you have added has improved the manuscript. However, I still have minor comments:
1. The title "Psychological and physiological responses in patients with generalized anxiety disorder: the use of acute exercise and virtual nature environment" should be revised. Authors are not only measuring a virtual nature environment but two types of VR scenarios. Does "virtual nature environment" mean either VN and VAP. Is an abstract painting considered a nature environment? If you are comparing both scenarios, the differences between them should be clear in the whole manuscript.
2. In the flow chart the N and name of the group at post-test needs to be revised: "VN group (n=40) & VN group (n=40)"
